# Enhancing the Deposition Rate and Uniformity in 3D Gold Microelectrode Arrays via Ultrasonic-Enhanced Template-Assisted Electrodeposition

**DOI:** 10.3390/s24041251

**Published:** 2024-02-15

**Authors:** Neeraj Yadav, Flavio Giacomozzi, Alessandro Cian, Damiano Giubertoni, Leandro Lorenzelli

**Affiliations:** 1Department of Industrial Engineering, University of Trento, 38123 Trento, Italy; 2Center for Sensors & Devices (SD), FBK—Foundation Bruno Kessler, 38123 Trento, Italy; giaco@fbk.eu (F.G.); acian@fbk.eu (A.C.); lorenzel@fbk.eu (L.L.)

**Keywords:** electrodeposition, MEA, uniformity, three-dimensional, template assisted, deposition rate

## Abstract

In the pursuit of refining the fabrication of three-dimensional (3D) microelectrode arrays (MEAs), this study investigates the application of ultrasonic vibrations in template-assisted electrodeposition. This was driven by the need to overcome limitations in the deposition rate and the height uniformity of microstructures developed using conventional electrodeposition methods, particularly in the field of in vitro electrophysiological investigations. This study employs a template-assisted electrodeposition approach coupled with ultrasonic vibrations to enhance the deposition process. The method involves utilizing a polymeric hard mask to define the shape of electrodeposited microstructures (i.e., micro-pillars). The results show that the integration of ultrasonic vibrations significantly increases the deposition rate by up to 5 times and substantially improves the uniformity in 3D MEAs. The key conclusion drawn is that ultrasonic-enhanced template-assisted electrodeposition emerges as a powerful technique and enables the development of 3D MEAs at a higher rate and with a superior uniformity. This advancement holds promising implications for the precision of selective electrodeposition applications and signifies a significant stride in developing micro- and nanofabrication methodologies for biomedical applications.

## 1. Introduction

Microelectrode arrays (MEAs) are powerful tools for studying the electrical activity in cells and tissues and consist of electrodes that record or stimulate electrical activity in vitro. Three-dimensional (3D) MEAs have become increasingly popular, as they provide a larger surface area for cell–electrode interactions and can capture more physiologically relevant data [1,2,3,4]. These MEAs are essential in various fields, particularly neuroscience and neuro-engineering, and serve as the foundation for groundbreaking technologies like brain–computer interfaces (BCIs). The efficacy of an MEA hinges on its sampling capabilities, which are determined by the electrode density and its capacity to precisely target specific regions of interest [5]. Although current fabrication techniques have made significant strides in enhancing the recording density, primarily through advancements in micro-electromechanical system (MEMS) fabrication methods [6,7,8], there are inherent limitations to existing technologies. Silicon-based arrays, for instance, have a limited volumetric electrode density and lack customization options. Similarly, alternative fabrication approaches such as bead stacking, 3D printing, and direct laser writing techniques provide options for individual shank customization and allow reproducibility; however, these techniques lack scalability in production and cost-effectiveness [9,10,11,12,13].

The forthcoming generation of electrophysiological recording tools must transcend these limitations and allow for the customization of probes to the specific study [14]. Historically, MEA fabrication methods have mirrored trends in the semiconductor industry, transitioning from micro-wires to lithography [15,16,17]. However, the emergence of template-assisted electrodeposition techniques presents a promising new avenue for the development of 3D MEAs with customizable shank heights and array topographies. Template-assisted electrodeposition utilizes a polymeric hard mask as a template to define the shape of the electrodeposited microstructures [18,19,20,21]. More recently, the development of multi-depth probing 3D MEAs with customizable microelectrode heights has been reported using the template-assisted electrodeposition technique [22]. Electrodeposited gold electrodes have a higher surface roughness compared to the electrodes developed using physical vapor deposition methods; this surface roughness translates to a larger electrochemically active area at the cell–electrode interface, promoting interactions between the cell and the electrode [4]. The cell–electrode adhesion could be improved further by functionalizing the electrode surface with adhesion promoters as a post-processing step [23].

The development of advanced 3D MEAs requires high electrodeposition rates to enable rapid prototyping and remain cost-effective, while the uniformity of the thickness of the electrodeposited microstructures plays a critical role in defining the array topography; however, conventional processes, particularly template-assisted electrode processes, suffer from low deposition rates due to localized depletion of ions in the electroplating solution, leading to a non-uniform deposition. The use of ultrasonic baths, which accelerate chemical reactions within liquid media using high-frequency pressure waves, appears to be very promising for overcoming the limitations of the conventional electrodeposition process. These baths are widely used in various industrial and laboratory applications, such as cleaning, degreasing, and electroplating [24,25]. In recent years, ultrasonic baths have also been proposed as a potential solution for accelerating the electrodeposition process, which is often limited by its low deposition rate. When an ultrasonic bath is used for electrodeposition, the sound waves create microscopic cavitation bubbles in the liquid medium. These bubbles are subjected to intense pressure and temperature changes, forming highly reactive sites on their surface. These sites can then act as catalysts, accelerating the electrodeposition process [26,27]. The potential applications of ultrasonic baths in electrodeposition are numerous. Ultrasonic baths could be used to produce gold microstructures with a high accuracy and uniformity. Additionally, using ultrasonic baths can reduce the cost of the electrodeposition process by increasing the deposition rate without needing an additional catalytic agent. While the effect of ultrasonic agitation on electrodeposition has been extensively studied for two-dimensional (2D) structures, its influence on template-assisted electrodeposition of 3D microstructures remains largely unexplored [28,29,30,31,32,33].

In this work, we investigate the influence of ultrasonic vibrations on the deposition rate and uniformity of microelectrode arrays developed using template-assisted electrodeposition. Various characterization techniques, including scanning electron microscopy, optical profilometry, and X-ray diffraction analysis (XRD), are used to assess the quality of the fabricated structures. A mechanical shear strength testing tool is utilized to test the adhesion strength of the electrodeposited micro-pillars with the planar substrate. The results of this study provide a deeper insight into the optimization of ultrasonic parameters for template-assisted electrodeposition, leading to an improved 3D MEA prototyping process. This study could also have broader implications for using ultrasonic agitation in other electrodeposition techniques for micro- and nanofabrication.

## 2. Materials and Methods

### 2.1. Materials

An additive-free electroplating solution (AUROLYTE CN200, Atotech Deutschland GmbH & Co. KG, Berlin, Germany) was chosen for the electrodeposition experiments. Three types of planar MEAs arranged in a hexagonal pattern (as described in [22]) were used for the experiments: (S1) consisting of 60 electrodes with a diameter of 65 µm and a pitch of 265 µm, of which 21 were connected to a custom routing for electrodeposition; (S2) consisting of 60 electrodes with a diameter of 65 µm and a pitch of 265 µm, of which 41 were connected to a custom routing for electrodeposition; and (S3) consisting of 60 electrodes with a diameter of 35 µm and a pitch of 195 µm, of which 44 were connected to a custom routing for electrodeposition. A thick negative-type photoresist (KMPR-1035, Kayaku Advanced Materials, Inc., Westborough, MA, USA) was utilized for template fabrication. An ultrasonic bath (Sonorex Digitec DT 514 BH-RC, BANDELIN electronic GmbH & Co. KG, Berlin, Germany) operating at a peak power of 640 W at 35 kHz was utilized for the experiments. The galvanostat used for these experiments was assembled in-house. Remover PG (Kayaku Advanced Materials, Inc., Westborough, MA, USA) was used to strip the photoresist template after completion of the electrodeposition process.

### 2.2. Template Development

Custom planar MEAs were designed and developed for this study, and the design and fabrication protocols are described in detail in Appendix C. The planar MEAs were first coated with a 110 µm-thick layer of KMPR photoresist. This was achieved by using a spin coater operating at 1000 rpm. This was followed by a 30 min soft-bake at a temperature of 100 °C. UV exposure was performed using an i-line mask aligner (KARL SÜSS MA6, SÜSS MICROTEC SE, Garching, Germany) to define the custom pattern on the photoresist. The photoresist was subjected to a post-exposure bake at 100 °C for a span of 6 min. This step was crucial to complete the curing reaction of the exposed regions. The final step was to develop the photoresist using the SU-8 developer solution supplied by MicroChemicals GmbH, Ulm, Germany. This 20 min process required a shaker plate for mild agitation for assistance. This process resulted in an array of 110 µm deep, 65 µm diameter cylindrical holes strategically aligned over the planar electrodes for S1- and S2-type MEAs, while the S3 MEA had cylindrical cavities with an internal diameter of 35 µm and a height of 110 µm.

### 2.3. Experimental Setup

All of the electrodeposition experiments were set up inside an ultrasonic bath with a temperature maintained at 55 °C, as illustrated in Figure 1. The electroplating solution bath was conditioned for 1 h before initiating gold electrodeposition for each experiment. The experiments were conducted in four phases. Experiment 1: Electrodeposition of gold MEAs with a high current density (i.e., 8 mA/cm^2^) for 1 h (H1) and various operational modes of the ultrasonic bath (i.e., no sonication mode (NS), pulsed sonication (PS) mode with a duty cycle of 50%, and continuous sonication (CS) mode) to investigate the influence of the ultrasonic bath operation mode on the deposition rate and electrode height uniformity across the array, as illustrated in Figure 1.

Experiment 2 (L1): This was a repeat of Experiment 1 with a lower current density (4 mA/cm^2^) to determine the influence of the deposition current density on the deposition rate and the uniformity of the electrode height. Experiment 3: The best electrodeposition parameters were selected from the previous experiments (i.e., CS-L1, based on the optimal deposition rate and the highest uniformity). In this case, the tests were repeated for longer durations of 2 and 3 h (i.e., CS-L2 and CS-L3, respectively) to verify the consistency of the process. Finally, a fourth experiment was conducted to verify the scalability of the process for the deposition of high-density (HD) MEAs for long durations. For the first two experiments, the S1 MEAs with 21 active electrodes were subjected to template-assisted electrodeposition for 1 h. For the third experiment, the S2 MEAs with 41 active electrodes were subjected to template-assisted electrodeposition for 2 and 3 h, and the experiment was repeated to ensure reproducibility. S3 MEAs were used in the fourth experiment by subjecting them to template-assisted electrodeposition for 4 h. The experimental details are tabulated in Table 1.

### 2.4. Analysis

A scanning electron microscope (SEM) was utilized to assess the process yield (defined as the number of electrodeposited micro-pillars divided by the number of electrodes subjected to electrodeposition for each MEA), and in order to determine the electrodeposition rate and uniformity, a two-step process was employed. First, an optical profilometer was used to measure the height of the electrodeposited micro-pillars, which were subjected to electrodeposition for one hour. To evaluate the height uniformity of the electrodeposited micro-pillars, the heights of multiple micro-pillars sourced from different regions of the MEA were analyzed comparatively.

To evaluate the mechanical, or more specifically, the shear strength, of the electrodeposited micro-pillars, a destructive die shear strength test was performed using the Condor Ez Pull&Shear test tool from XYZtec, as shown in Figure 2a.

For each experimental condition from the first experiment, five micro-pillars randomly selected from each MEA were subjected to a shear test at room temperature. A shear tool, moving at a constant velocity of 2 µm/s, was placed in contact with the side of the micro-pillar until fracture or split occurred, as illustrated in Figure 2b. To obtain a valid measure of the adhesion strength at the pillar–substrate interface, it was ensured that failure occurred at the pillar–substrate interface by placing the shear tool 1 µm above the substrate surface. The maximum force applied by the tool at the point of failure was recorded for all five measurements. From the mean maximum shear force and its standard deviation, the maximum shear strength *τ*_max_ was calculated from the maximum shear force *F*_max_ using the following equation:*F*_max_ = *τ*_max_ · *A*(1)
where *A* is the cross-sectional area of the pillar, obtained from the measured pillar diameter.

In order to evaluate the influence of ultrasonic vibrations on the morphological characteristics of the electrodeposited gold microstructures, three separate samples were prepared. Square (17 × 17 mm^2^) glass wafer pieces coated with 5 nm chromium (Cr) and 200 nm gold (Au) deposited via the thermal evaporation technique were utilized as substrates for this experiment. The substrates were masked with 70 µm-thick Kapton tape consisting of a circular opening with a diameter of 6 mm. The masked substrates were subjected to electrodeposition using the parameters from the first experiment. The first substrate was subjected to electrodeposition for 1 h in the absence of ultrasonic vibrations, i.e., NSED. The second substrate was subjected to electrodeposition with an ultrasonic bath operating in pulsed sonication mode for one hour, i.e., PSED. Finally, the third substrate was subjected to 1 h of electrodeposition under continuous sonication mode, i.e., CSED, as shown in Figure 3. The influence of ultrasonic vibrations on the morphology of the electrodeposited gold films was determined by measuring the surface roughness using atomic force microscopy (AFM; PX, NT-MDT SI, Moscow, Russia).

The evaluation of the influence of ultrasonic vibrations on the structural (crystal phase) characteristics of the electrodeposited structures was carried out via X-ray diffraction (XRD) on the NSED, PSED, and CSED samples and the substrate with the seed layer using a high-resolution XRD instrument (ITALSTRUCTURES APD2000, Austin AI, Austin, TX, USA). The average grain size *D* was determined via XRD for multiple crystallographic planes, including (111), (311), (220), and (200), via Scherrer’s equation:(2)D=K·λ/(β·cos(θ))
where the Scherrer constant *K* is typically taken as 0.94 [34], *λ* = 1.5418 Å is the wavelength of the Cu Kalpha radiation employed for the XRD, *β* is the full-width at half-maximum (FWHM) of the measured peak in radians, and theta is the Bragg angle at which the peak occurs.

## 3. Results and Discussions

### 3.1. Deposition Rate and Uniformity

The deposition rate and uniformity are critical attributes in the template-assisted electrodeposition process, and they play a pivotal role in determining the quality of the micro-pillars. An optical profilometer was utilized to measure the height of the electrodeposited micro-pillars across each array to derive the deposition rate and assess the uniformity of the height of the electrodeposited micro-pillars. In the first experiment, the S1 MEAs were subjected to electrodeposition for 1 h with a high current density of 8 mA/cm^2^ (H1) under different operational modes of the ultrasonic bath. The MEA subjected to electrodeposition without ultrasonic vibrations (i.e., NS-H1) showed a deposition rate of 0.10 µm/min and a percentage standard deviation of 35.8%. This indicates a low deposition rate compared to standard electrodeposition without the use of a template under similar experimental conditions (i.e., ~0.45 µm/min) and a considerable variation in micro-pillar heights across the array, revealing a lack of uniformity. The MEA subjected to electrodeposition with pulsed ultrasonic vibrations (i.e., PS-H1) showed an improved deposition rate of 0.24 µm/min. The percentage standard deviation decreased to 16.19%, suggesting an improved uniformity compared to NS-H1. Finally, the MEA subjected to electrodeposition with continuous ultrasonic vibrations (i.e., CS-H1) demonstrated the highest deposition rate of 0.55 µm/min. It also showed a further decrease in the percentage standard deviation to 13.52%, implying more consistency in micro-pillar heights and an enhanced uniformity. The normalized height distributions of electrodeposited microstructures (micro-pillars) for each sample from individual experiments are plotted in the bar graphs in Appendix B.

In the second experiment, the S1 MEAs were subjected to electrodeposition for 1 h with a lower current density of 4 mA/cm^2^ (L1) under different operational modes of the ultrasonic bath. The MEA subjected to electrodeposition without ultrasonic vibrations (i.e., NS-L1) showed a deposition rate of 0.06 µm/min proportional to NS-H1 and a percentage standard deviation of 14.77%, indicating a considerable improvement in the uniformity. The MEA subjected to electrodeposition with pulsed ultrasonic vibrations (i.e., PS-L1) showed an improved deposition rate of 0.13 µm/min compared to NS-L1. However, the experiment revealed a slight increase in the percentage standard deviation (i.e., 16.15%) with respect to NS-L1. Finally, the MEA subjected to electrodeposition with continuous ultrasonic vibrations and lower deposition current (i.e., CS-L1) showed a deposition rate of 0.24 µm/min and also a further decrease in the percentage standard deviation to 9.63%, indicating more consistency in the micro-pillar heights, and therefore an improved uniformity.

The first experiment clearly indicated that the use of ultrasonic vibrations significantly increases the deposition rate and improves uniformity. A higher deposition rate is always a desirable parameter; however, uniformity is an essential parameter for developing MEAs. The second experiment demonstrated a reduction in the deposition rate compared to the first experiment but this was proportional to the applied deposition current density in all three cases, as shown in Figure 4a. The CS-L1 showed the lowest percentage standard deviation, indicating the highest uniformity compared to all the other samples from the first and second experiments, as shown in Figure 4b. Based on these results, continuous sonication with a low current density was chosen as the optimal combination for further investigations.

In the third experiment, S2 MEAs consisting of 41 active electrodes were subject to template-assisted electrodeposition for 2 and 3 h using continuous sonication with a low current density (i.e., CS-L2 and C3-L3, respectively). The third experiment was repeated to ensure reproducibility (i.e., CS-L2 r and C3-L3 r). Both CS-L2 and the CS-L3 have comparable but slightly lower deposition rates (CS-L3 having the lowest deposition rate) compared to the CS-L1 experiment, and the trend was confirmed when the experiment was repeated, i.e., CS-L2 r and C3-L3 r, as shown in Figure 4c. This decrease in the deposition rate during the longer deposition is likely due to the depletion in gold ions in the electroplating solution, However, this is mere speculation based on the experimental setup; further investigations are warranted to fully understand this observation. On the other hand, the percentage standard deviation decreased to as low as 2.13% for the CS-L3 experiment as the duration of the electrodeposition increased, as shown in Figure 4d.

Finally, the fourth experiment was conducted by subjecting an HD MEA (S3) to template-assisted electrodeposition for 4 h (i.e., CS-L4 (HD)) with continuous ultrasonic vibrations and a current density of 4 mA/cm^2^. The experiment led to a deposition rate of 0.24 µm/min, consistent with the previous experiments, as shown in Figure 4c. The experiment resulted in the lowest standard deviation of 1.76%, indicating a very high uniformity, as shown in Figure 4d.

These observations suggest that employing ultrasonic assistance, especially in continuous mode, substantially boosts the deposition rates in template-assisted electrodeposition. Moreover, the lower deposition current density significantly enhances the uniformity in the electrode height across the array, as shown in Figure 5. The empty circular sites within the images in Figure 5 are the planar electrodes that were not subjected to electrodeposition. All the MEAs subjected to electrodeposition demonstrated a 100% deposition yield (i.e., the number of electrodeposited micro-pillars divided by the number of electrodes subjected to electrodeposition). The data for the micro-pillar heights are available in Appendix A. This finding may provide a crucial strategy for optimizing the deposition parameters when precise control over micro-pillar dimensions and uniformity is desired.

### 3.2. Mechanical Strength

The mechanical strength of the electrodeposited micro-pillars was a crucial parameter evaluated in this study to investigate the influence of ultrasonic vibrations on the shear strength of the electrodeposited micro-pillars. To this end, a destructive shear stress test was employed to determine the maximum shear force the micro-pillars could endure before failure, as shown in Figure 2. The three MEAs (five micro-pillars from each MEA) from the first experiment and the HD MEA from the fourth experiment were subjected to this test.

For the NS-H1 MEA, the maximum shear force was recorded at 0.610 N with a percentage standard deviation of 6.58%. The PS-H1 MEA displayed a slightly higher resistance, having a maximum shear force of 0.707 N and a lower percentage standard deviation of 3.57%, suggesting a marginally superior and more consistent shear strength compared to NS-H1. The CS-H1 MEA documented a mean maximum shear force similar to NS-H1, at 0.636 N, but with a decreased percentage standard deviation of 3.38%, hinting at a more uniform shear strength under this process.

The CS-L4 (HD) MEA registered a substantially lower shear force resistance at 0.158 N with a percentage standard deviation of 3.62%. However, it is imperative to note that the cross-sectional area of the micro-pillars from the CS-L4 (HD) MEA was 3.5 times smaller than the micro-pillars from the other three methods. As shear strength is directly proportional to the cross-sectional area, the seemingly weaker shear strength in the CS-L4 (HD) MEA is not an indication of a poor shear resistance but a consequence of a smaller cross-sectional area. Adjusting for the reduced cross-sectional area, the actual shear strength (*τ*_max_) of the CS-L4 (HD) MEA is similar to the CS-H1 MEA, as shown in Figure 6a. This observation suggests that the reduced current density in continuous ultrasonic bath-assisted electrodeposition with a lower deposition rate does not necessarily undermine the shear strength. The findings from this research highlight the impact of ultrasonic agitation and the current density on shear strength, having potential implications for the design of electrodeposition processes for gold microstructures that require strong resistance to shear forces on the substrate. The data pertaining to the shear strength measurements are presented in Appendix A. Figure 6b shows an optical image of the shear strength measurement setup with the contact tool at the initial measurement position for the CS-L4 (HD) MEA.

### 3.3. Structural and Morphological Analysis

Figure 7 shows the Y-stacked plot of the XRD spectra for the three electrodeposited gold samples (i.e., NSED, PSED, and CSED) and the substrate with the gold seed layer (substrate) deposited via the thermal evaporation technique. As expected, the seed layer has a crystalline structure with a single peak at 38.2° (i.e., the (111) plane). In the case of the electrodeposited samples, strong peaks were observed in the (111) plane, and weaker peaks were observed in the (311), (220), and (200) planes, indicating a polycrystalline structure. No significant shift in the 2theta locations in the XRD spectra was observed for the various electrodeposited gold samples. As shown in Table 2, the full width at half maximum (FWHM) of the (111), (311), (220), and (200) planes is highest for the NSED sample and lowest for the PSED sample, indicating a lower surface roughness and average gain size compared to the PSED and CSED samples, which could also be observed in the AFM images in Figure 8.

### 3.4. Implications for 3D MEAs

These results demonstrate that ultrasonic-bath-assisted electrodeposition, especially under continuous operation (CSED and LC-CSED), significantly improves the performance of template-assisted electrodeposition. This could allow for the fabrication of higher quality 3D MEAs, as it leads to a higher deposition rate, improved uniformity, and an enhanced adhesion strength of gold micro-pillars. The findings could also guide the optimization of ultrasonic parameters in electrodeposition processes, advancing the field of micro- and nanofabrication. Our results highlight the promising potential of ultrasonic-bath-assisted electrodeposition for fabricating 3D MEAs. Future research can further validate these findings by exploring other factors that influence the electrodeposition process and optimizing them for superior MEA performance.

## 4. Conclusions

This research explored the impact of ultrasonic vibrations on template-assisted electrodeposition of gold micro-pillars for the development of 3D MEAs intended for in vitro electrophysiological investigations. This study found that continuous ultrasonic-bath-assisted electrodeposition, at both high and low deposition current densities, significantly enhanced the deposition rate and improved the thickness/height uniformity of the micro-pillars across the array, specifically for template-assisted electrodeposition processes. Along with the application of continuous ultrasonic vibrations, the deposition time and current density also play a crucial role in improving uniformity. The direct relationship between the current density and the grain size is well established. However, the observed relationship between the electrodeposition duration and uniformity presents an interesting opportunity for further investigation. The investigation outcomes are crucial in understanding how to optimize the use of ultrasonic baths in template-assisted electrodeposition, thereby improving the fabrication of 3D MEAs. These results therefore significantly contribute to the advancement of micro- and nanofabrication. These findings could serve as a valuable guide for optimizing the parameters of ultrasonic-bath-assisted electrodeposition.

While the study’s results offer exciting possibilities for ultrasonic-assisted electrodeposition, further investigations are warranted. These could include investigations of different materials for electrodeposition, alternative ultrasonic setups, and a wider range of ultrasonic parameters. Also, further investigations of how these findings translate into real-world applications in biomedical devices and MEMSs would be beneficial. The research outcomes, nevertheless, offer promising avenues to explore and optimize ultrasonic-assisted electrodeposition methods for the fabrication of 3D MEAs, bringing us a step closer to achieving high-performance biomedical interfaces.

## Figures and Tables

**Figure 1 sensors-24-01251-f001:**
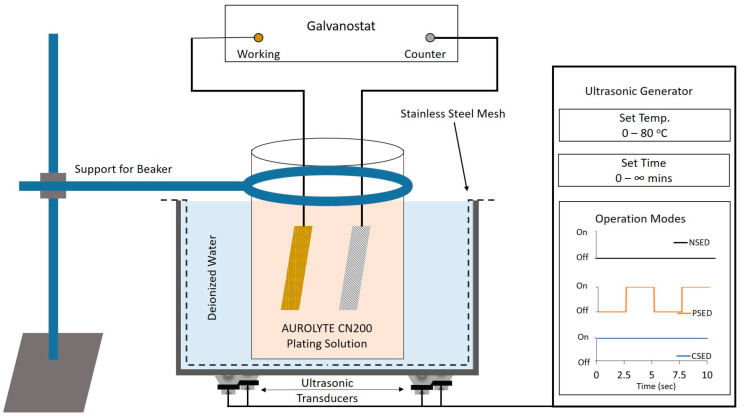
Schematic representation of the experimental setup including the various operational modes of the ultrasonic bath, i.e., NSED (electrodeposition without the ultrasonic vibrations), PSED (electrodeposition with ultrasonic vibrations in pulsed mode with a duty cycle of 50%), and CSED (electrodeposition with continuous ultrasonic vibrations).

**Figure 2 sensors-24-01251-f002:**
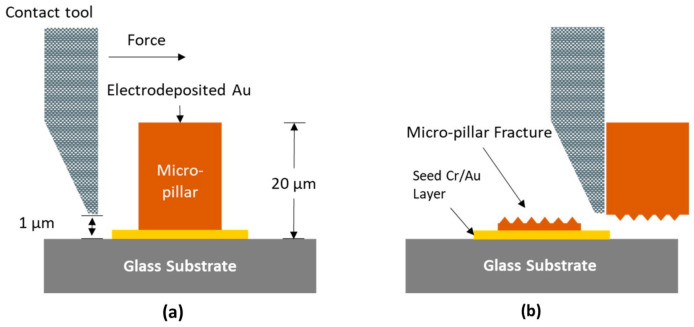
Illustration of the shear test setup. (**a**) Initial position of the contact tool, (**b**) final position of the contact tool.

**Figure 3 sensors-24-01251-f003:**
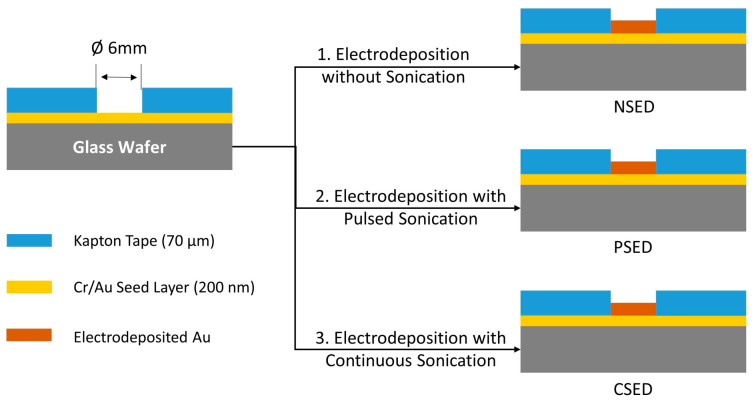
Schematic representation for preparation of the samples for the structural and morphological characterizations.

**Figure 4 sensors-24-01251-f004:**
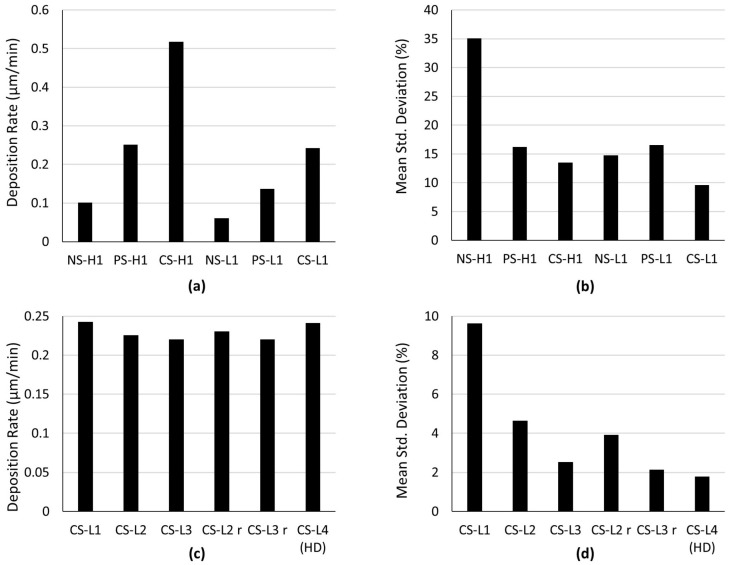
Bar graphs comparing the deposition rate and the percentage mean standard deviation across the height of the electrodeposited micro-pillar array (as a measure of uniformity) for the various experiments. (**a**) Comparison of the deposition rate of the different microelectrode arrays (MEAs) from the first (H1) and the second (L1) experiment. (**b**) Comparison of the percentage mean standard deviation in the thickness of 21 electrodeposited micro-pillars for each MEA in the first and second experiment. (**c**) Comparison of the deposition rate of the various microelectrode arrays (MEAs) from the second (CS-L1), third (CS-L2, CS-L3, CS-L2 r, and CS-L3 r), and fourth (CS-L4 (HD)) experiments. (**d**) Percentage mean standard deviation pertaining to each sample from the second (CS-L1), third (CS-L2, CS-L3, CS-L2 r, and CS-L3 r), and fourth (CS-L4 (HD)) experiments.

**Figure 5 sensors-24-01251-f005:**
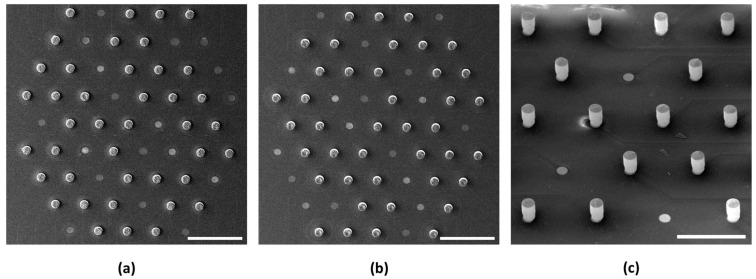
SEM images of the MEAs. (**a**) CS-L2 with a tilt of 10 degrees (scale: 500 µm). (**b**) CS-L3 with a tilt of 10 degrees (scale: 500 µm). (**c**) CS-L4 (HD) with a tilt of 25 degrees (scale: 200 µm).

**Figure 6 sensors-24-01251-f006:**
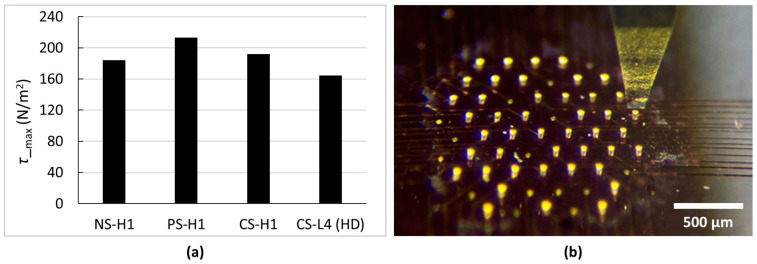
Shear strength analysis. (**a**) Bar chart presenting the average (n = 5) of the maximum shear strength *τ*_max_ faced by the micro-pillars before failure for each MEA subjected to the test. (**b**) A microscopic image of the CS-L4 (HD) MEA undergoing the shear test with the contact tool at the initial position.

**Figure 7 sensors-24-01251-f007:**
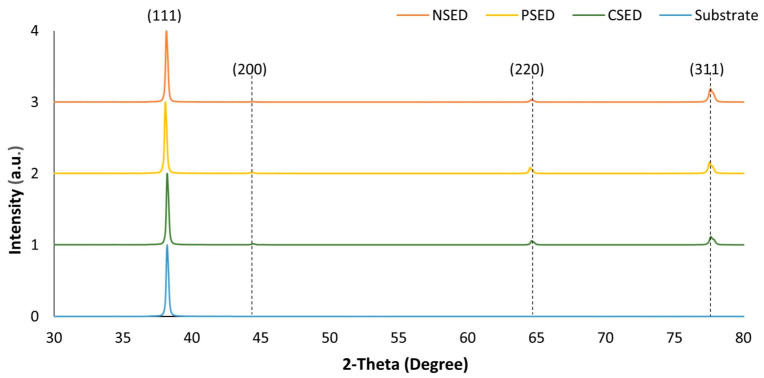
Measured XRD patterns of the electrodeposited gold (without template) under different ultrasonic vibration modes and the thermally evaporated gold seed layer (i.e., substrate).

**Figure 8 sensors-24-01251-f008:**
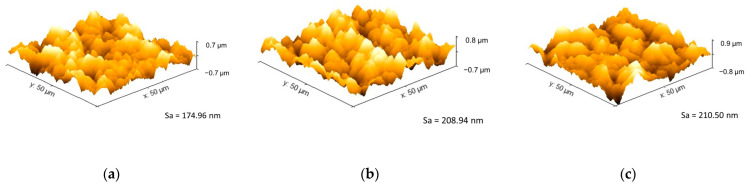
AFM 3D image and average surface roughness (Sa) of (**a**) NSED, (**b**) PSED, and (**c**) CSED.

**Table 1 sensors-24-01251-t001:** Design of experiments. List of experiments performed under various conditions.

	Sample Name	MEA Type	No. of Active Electrodes	Electrode Diameter (µm)	Deposition Current Density (mA/cm^2^)	Electrodeposition Duration (Minutes)	Ultrasonic Bath Mode *
Experiment 1	NS-H1	S1	21	65	8	60	NS
	PS-H1	S1	21	65	8	60	PS
	CS-H1	S1	21	65	8	60	CS
Experiment 2	NS-L1	S1	21	65	4	60	NS
	PS-L1	S1	21	65	4	60	PS
	CS-L1	S1	21	65	4	60	CS
Experiment 3	CS-L2	S2	41	65	4	120	CS
	CS-L3	S2	41	65	4	180	CS
	CS-L2 r **	S2	41	65	4	120	CS
	CS-L3 r	S2	41	65	4	180	CS
Experiment 4	CS-L4 (HD)	S3	44	65	4	240	CS

* Ultrasonic bath modes: NS—ultrasonic vibrations OFF, PS—ultrasonic vibrations ON in pulsed mode, and CS—ultrasonic vibrations ON continuously. ** Repeated.

**Table 2 sensors-24-01251-t002:** Full width at half maximum (FWHM) of 111, 311, 220, and 200 peaks of electrodeposited gold.

Sample	(111) ^1^	(311)	(220)	(200)	Average Grain Size (nm)
	FWHM	FWHM	FWHM	FWHM	
Substrate	0.20524	-	-	-	40.96
NSED	0.15433	0.25735	0.21134	0.23835	43.63
PSED	0.14565	0.2244	0.1736	0.23286	48.51
CSED	0.14751	0.23751	0.18359	0.2303	47.01

^1^ Crystallographic plane.

## Data Availability

The original contributions presented in the study are included in the article/Appendix A, further inquiries can be directed to the corresponding author/s.

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
