# Peer review of "Enhancing the Deposition Rate and Uniformity in 3D Gold Microelectrode Arrays via Ultrasonic-Enhanced Template-Assisted Electrodeposition"

_sensors, 2024, doi:10.3390/s24041251_

Round 1

Reviewer 1 Report (Previous Reviewer 1)

Comments and Suggestions for Authors

The proposed methods is useful in the process of 3D MEA fabrication.  However only one plating solution was demonstrated in the manuscript. If the author can not prove the methods is effective for other materials, the title of the manuscript had better be defined.

Author Response

Reviewer 2 Report (Previous Reviewer 2)

Comments and Suggestions for Authors

I believe that authors have met all the requirements and answered all my comments, I now recommend the manuscript for publications in 'Sensors'.

Author Response

Reviewer 3 Report (Previous Reviewer 3)

Comments and Suggestions for Authors

I have two pending comments/experiments that must be addressed before I can recommend this manuscript for publication.

My primary concern is that these electrodes have not been tested for their electrical properties, despite being designed for measurement systems. It is crucial to investigate the resistance of the pillars and the capacitance of the interface, as these may be affected by rapid deposition-induced morphological changes. To address this, these devices, which have been fabricated both with ultrasound-enhanced deposition and conventional electrodeposition methods, should be utilized for impedance measurements in an in vitro setting. The impedance measurements should be carefully analyzed, and the data should be used to build an RC circuit model. Any observed decay in the values of each component should be thoroughly documented.

Authors suggest ‘When an ultrasonic bath is used for electrodeposition, the sound waves create microscopic cavitation bubbles in the liquid medium. These bubbles are subjected to intense pressure and temperature changes, forming highly reactive sites on their surface. These sites can then act as catalysts, accelerating the electrodeposition process’. Based on the authors' findings, it is evident that the use of ultrasound improves the deposition rate. However, I still have doubts about the uniformity. Does the accelerated deposition not result in the depletion of Au nanoparticles near the deposition site, potentially leading to inhomogeneity? For example, when comparing the uniformity achieved with pulsed ultrasound waves to that with continuous ultrasound, there is a notable reduction in uniformity. I would expect the opposite outcome, given that the intervals between pulses could potentially allow for the diffusion of gold particles to the active sites.  Furthermore, it is essential to explore the uniformity by generating two electrodes, one with a higher electrodeposition current density (mA/cm2) without ultrasound and one with a lower electrodeposition current density (mA/cm2) with ultrasound, while keeping the total deposition rate (µm/min) constant. This experiment would give a definitive answer on relationship between uniformity and deposition rate using each modality. Elaborating on this phenomenon in more detail is crucial for the development of a dependable protocol and for discussing potential drawbacks and considerations associated with each approach.

Author Response

Reviewer 4 Report (New Reviewer)

Comments and Suggestions for Authors

The research article by N.Yadav et al. investigates the impact of different ultrasonic vibrations in template-assisted electrodeposition of gold micro-pillars; these types of structures constitute the conductive elements in 3D MEA devices, used especially for in vitro electrophysiological applications. The results clearly demonstrate that ultrasonic vibration, especially in continuous mode, improves the electrodeposition leading to a higher deposition rate, better uniformity and improved adhesion strength of micro-pillars. The experimental procedures are systematic and supported by the statistics. Figures are clear and the results are interesting with a high potential of disruptive applications. However, some points need to be addressed while other should be further discussed before considering, from my side, the manuscript suitable for Sensors. Below, the list of points need to be addressed

- At lines 243-245 authors state “However, the experiment showed a significant increase in percentage standard deviation (i.e. 16.15 %) w.r.t. NS-L1 and PS-H1.” From the graph in Figure 4, the increment in percentage standard deviation seems to be not so significant, especially with respect to PS-H1 where there is no increment (16.19% vs 16.15%). Please, check it.

-At lines 266-268 authors state “This decrease in deposition rate during the longer deposition time is most likely due to the depletion of gold ions in the electroplating solution.” Can the authors justify this claim? You could add references about it and implement the discussion even with slight speculations. This could increase the manuscript readability.

-At lines 280-282 authors state “These observations suggest that employing ultrasonic assistance, especially in continuous mode, substantially boosts the deposition rates in template-assisted electrodeposition.” The experiments conceptualization rightly highlighted the role of ultrasonic vibrations but the key function of the deposition time and of the current density should be emphasized as well.

-At lines 317-319 authors state “Adjusting for the reduced cross-sectional area, the actual shear  strength (τ_max) of the CS-L4 (HD) MEA is similar to, if not better than, the CS-H1 MEA, as shown in Figure 6a.” Figure 6a show a quite similar τ max for CS-L4 (HD) MEA respect to CS-H1, but no better, otherwise it would be necessary to specify what is meant by better. Please check it. Moreover, can the authors discuss about the cross-sectional area smaller than one obtained with other methods? This aspect could be crucial for the development of template assisted electrodeposition

- In the introduction section, the authors present an exhaustive state of the art on MEA devices including its fields of application and indicate the “capacity to precisely target specific regions of interest” as one of the main benefits of such a technology. For completeness, authors should mention as well the case of post-processing chemical functionalization to control the precision targeting of neural cells (http://dx.doi.org/10.1021/acs.langmuir.6b01352 ). This could enhance the manuscript attractiveness and draw interest of a larger scientific community including surface scientists, biophysicists and chemists.

Minor:

Figure 2: labels (a) and (b) are missing and the writing “Glass Substrate” should be on both the substrates otherwise they seems different; please add it.

Round 2

Reviewer 4 Report (New Reviewer)

Comments and Suggestions for Authors

The authors addressed all the points raised up in my previous round revision.

This manuscript is a resubmission of an earlier submission. The following is a list of the peer review reports and author responses from that submission.

Round 1

Reviewer 1 Report

Comments and Suggestions for Authors

The manuscript reports a kind of electrodeposition method enhanced by ultrasonic processing. The result was verified with a few of micro pillars which are fabricated on a public metal film with pattern models made of photoresist. By comparing the deposition rates and mechanical characteristics of the pillars fabricated with different parameters, the manuscript concludes that the proposed methods is useful.

  The application of the proposed methods, according to text written in the abstract and conclusion (“Driven by the need to overcome limitations in deposition rate and the height uniformity of the electrodeposited microstructures, particularly in the realm of neuroscientific research and brain-computer interfaces”,), is suit for neural electrode arrays (MEA). However the MEA uses for neural recording has: the lateral dimension is commonly no more than 50 μm, the height or length dimension is at least more than 1 mm. however the micro-pillar demonstrated in the text is 65 μm wide and several hundred micrometers height.  What’s more, MEA a composed of a series micro pillar of lines separately. The proposed method fabricated all the pillars on a public metal substrate. This kind of device can not be used to act as a MEAs.

Reviewer 2 Report

Comments and Suggestions for Authors

Authors investigated the influence of ultrasonic vibrations on the deposition rate, and uniformity of microelectrode arrays that were developed using template-assisted electrodeposition. While, the structure of the study seems adequate, I have the following concerns:

1.    I suggest a native English speaker or an editing service to through the entire manuscript.

2.    I suggest having the abstract free of cited references; abstract should focus on the presented work not the work of others. 

3.    Some of the figures that present results lake statistical data such as Figures 6(a), A1, A2, A3, A4 and A5. 

4.    While the experimental work is overwhelming, more in-depth analysis of the results is essentially required. 

I would recommend authors to take care of the above mentioned issues in the current manuscript and then resubmit a revised draft.

Comments on the Quality of English Language

1.  I suggest a native English speaker or an editing service to through the                 entire manuscript.

Reviewer 3 Report

Comments and Suggestions for Authors

In this work, the authors aim to build 3D electrodes that are necessary for enhancing neural interfacing. Current challenges for existing fabrication methodologies based on electrodeposition to build such structures are lack of homogeneity in pillar sizes and low deposition rates. To address these challenges, the authors employ ultrasound-assisted electrodeposition, successfully showcasing high aspect ratio 3D electrodes with improved uniformity and accelerated deposition rates. Overall I am in favor of this publication as it provides a comprehensive look on this method and its characterization across various parameters. However, I believe there are certain minor changes and clarifications required;

1) The primary objective of these devices, as extensively outlined in the introduction, is their application in the neural interfaces. Did the authors characterize the electrical properties of pillars in each case and how it changes as a function of each method in terms of conductivity and interface impedance? I believe it is essential to extensively characterize and show how recording properties of these devices will be affected by this particular deposition process -at least in saline solution. 

2) Main limitation in the homogeneity of conventional electrodeposition is the depletion of active particles near the electrodes during deposition. Authors note that the ultrasound-enhanced deposition methodology provides improvements due to creation of cavitation enhanced reactive sites in the surface of electrodes. In this case, while it is evident that ultrasound increases deposition rate, why should it also increase homogeneity? Does ultrasound not lead to faster depletion of active particles in proximity of the electrodes due to increased deposition rate? Alternatively, does ultrasound also boost the flux of active species in the solution to effectively replenish the concentration?

3) Did the authors utilize different frequency pulses in their study? While I understand that the upper limit may be constrained by the capabilities of the tool, it is crucial to mention how varying power or frequency modes could impact the deposition process for future research. Additionally, providing a comprehensive explanation of the principles of ultrasound physics –within the context of this research- would be valuable in order to comprehend their rationale for selecting this specific power level.